# The Oral Health of Refugees and Asylum Seekers in Canada: A Mixed Methods Study Protocol

**DOI:** 10.3390/ijerph16040542

**Published:** 2019-02-13

**Authors:** Mary Ellen Macdonald, Mark T. Keboa, Nazik M. Nurelhuda, Herenia P. Lawrence, Franco Carnevale, Mary McNally, Sonica Singhal, Khady Ka, Belinda Nicolau

**Affiliations:** 1Faculty of Dentistry, McGill University, 500-2001 McGill College, Montréal, QC H3A 1G1, Canada; mark.keboa@mail.mcgill.ca (M.T.K.); khady.ka@canada.ca (K.K.); belinda.nicolau@mcgill.ca (B.N.); 2Faculty of Dentistry, University of Toronto, 124 Edward St, Toronto, ON M5G 1G6, Canada; nazik.suleiman@dentistry.utoronto.ca (N.M.N.); herenia.lawrence@dentistry.utoronto.ca (H.P.L.); sonica.singhal@oahpp.ca (S.S.); 3Ingram School of Nursing, McGill University, 680 Sherbrooke West 1800, Montréal, QC H3A 2M7, Canada; franco.carnevale@mcgill.ca; 4Faculty of Dentistry, Dalhousie University, 5981 University Avenue, Halifax, NS B3H 4R2, Canada; mary.mcnally@dal.ca

**Keywords:** refugees, migrants, oral health, mixed methods design, health policy

## Abstract

Canada received over 140,000 refugees and asylum seekers between 2015 and 2017. This paper presents a protocol with the purpose of generating robust baseline data on the oral health of this population and build a long-term program of research to improve their access to dental care in Canada. The three-phase project uses a sequential mixed methods design, with the Behavioral Model for Vulnerable Populations as the conceptual framework. In Phase 1a, we will conduct five focus groups (six to eight participants per group) in community organizations in Ontario, Canada, to collect additional sociocultural data for the research program. In Phase 1b, we will use respondent-driven sampling to recruit 420 humanitarian migrants in Ontario and Quebec. Participants will complete a questionnaire capturing socio-demographic information, perceived general health, diet, smoking, oral care habits, oral symptoms, and satisfaction with oral health. They will then undergo dental examination for caries experience, periodontal health, oral pain, and traumatic dental injuries. In Phase 2, we will bring together all qualitative and quantitative results by means of a mixed methods matrix. Finally, in Phase 3, we will hold a one-day meeting with policy makers, dentists, and community leaders to refine interpretations and begin designing future oral health interventions for this population.

## 1. Introduction

According to the United Nations (UN), there were 25.4 million refugees and 3.1 million asylum seekers (humanitarian migrants) globally by the end of 2017 [1]. Canada actively solicits these “humanitarian migrants” for resettlement: 82,010 refugees were resettled in Canada between 2015 and 2017, with an additional 58,320 requesting asylum [2]. Canada is a signatory on numerous UN conventions that welcome and attend to the health and wellbeing of humanitarian migrants [2]. In 2008, the World Health Assembly adopted a resolution calling on member states to improve the health of migrants [3]. These treaties morally oblige host countries to attend to the wellbeing of this population through health and social services policy.

Humanitarian migrants often arrive in their host countries with poor health, limited finances, linguistic challenges, and greatly-reduced family and community support. They are severely disadvantaged when entering a new sociocultural and political system, and vulnerable to multiple health risks [4]. To facilitate integration, wellbeing, and productivity, humanitarian migrants need a healthy start to maximize life in the host country [5,6].

While humanitarian migrants often arrive after harried journeys, potentially with months or years in refugee camps and poor overall health, there is little information on their oral health status or experiences seeking oral health care [7,8]. Poor oral health manifests in pain, poor nutrition, social stigma, and low quality of life; it leads both to and from serious systemic illnesses [9]. The stress endemic to the refugee experience elevates risk for oral health conditions, which further elevates risk of systemic diseases [10]. According to the World Health Organization (WHO), the lack of information on the oral health status, knowledge, and care seeking of humanitarian migrants constitutes an urgent knowledge gap [3].

Once in Canada, oral health care can be experienced as an expensive service. Data suggest that the general [9] and oral health [4,11] of immigrants declines in the first years of settlement. We can hypothesize a similar outcome for oral health of humanitarian migrants, yet to date this has not been studied in adults [4,9,12]. High-quality data are needed to help service providers and policy makers understand both individual and system level barriers and facilitators to their oral health (e.g., sociocultural [13], religious [10,14]) in order to design appropriate policy and services [15].

### 1.1. Research Questions

Our research protocol attends to this gap, answering the overarching research question: What is the oral health status of newly arrived adult humanitarian migrants and how can oral health services better meet their needs? We have four sub-questions:**Research Question (RQ) 1:***How do newly arrived humanitarian migrants understand/perceive oral health?***RQ2:***Through what pathways do newly arrived humanitarian migrants access oral health care?***RQ3:***What is the oral health status of newly arrived humanitarian migrants? We hypothesize a significantly higher proportion of humanitarian migrants will have untreated dental caries compared to Canadians.***RQ4:***What are the barriers to accessing oral health services among newly arrived humanitarian migrants?*

In answering these questions, we will be collecting baseline data on the oral health status of newly arrived humanitarian migrants in Canada. We will compare these data with an existing representative sample of Canadians and will identify factors associated with limited access to services. In so doing, this research will be the first to generate the foundational data needed to build a long-term program of research to improve access to dental care for this vulnerable Canadian population.

### 1.2. Frameworks

Our main conceptual framework, the Behavioral Model for Vulnerable Populations (BMVP) [16], proposes the characteristics of a population that determine use of health services. This model was chosen because of its successful use in oral health studies highlighting the domains that determine oral health status and use of dental services by vulnerable populations [17,18]. Domains include the following: (1) Predisposing factors: traditional variables (demographics); (2) Enabling factors: personal and family resources; (3) Need factors: perceived health and evaluated health; (4) Health behaviour: personal health practices; (5) Outcome factors: health status (perceived health, evaluated health) and satisfaction with care.

We have developed our questionnaire with questions from these BMVP domains; specific variables from these domains (e.g., perceived need for oral care, past dental experience) will be used in our analysis (see below).

In addition, the Canadian Health Measures Survey (CHMS) (which includes an oral health examination) and the Canadian Community Health Survey (CCHS) were both used to develop our questionnaire. The CHMS contains a validated oral health module using a standardized clinical and household protocol based upon WHO standards [19]. The CCHS is a national survey with three core oral health questions and 13 optional ones, adaptable to the needs of each health region in the country. 

We have also included the construct Oral Health Literacy (OHL) [20] in our questionnaire. OHL is the degree to which individuals have the capacity to obtain, process, and understand basic oral health information and services needed to make, and act on, health decisions [20]. We have included four screening questions from the OHL tool developed and evaluated for Canadian adults [21].

Finally, our qualitative component (ongoing) employs two main frameworks: the McGill Illness Narrative Interview (MINI) [21] and the BMVP [16]. The MINI is a theoretically based semi-structured interview guide designed to capture narratives of illness experience including the shared cultural aspects and behaviours of an individual or a group. These questions are divided into four sections: initial illness narrative; explanatory model; services and response to treatment; and impact on life. Using this tool therefore enables researchers to obtain an in-depth understanding of the participants’ illness experience [21]. The two frameworks were used to construct the interview instrument, as well as to conduct the thematic analysis.

### 1.3. Preliminary Work

In 2017, we conducted a qualitative study (Phase 1a) with a maximum variation sample of 25 humanitarian migrants and 10 healthcare providers in Montreal, Quebec. Interviews were recorded in participants’ preferred language, guided by the McGill Illness Narrative Interview [21] and the BMVP [16]. Domains of inquiry included the following: migration experience; experience of oral health and treatment; cultural understanding of oral health, hygiene, and care; care-seeking pathways; and barriers and facilitators to care. Participant-observation included data from community settings and mobile dental clinics to enhance sociocultural analysis. Iterative analysis was deductive, inductive, and contextual. Results indicated that participants were knowledgeable on causes of oral disease, the importance of oral health for general health, and ways to improve access to oral care. The restrictive Interim Federal Health Program (IFHP) policy, financial strain, and waiting times impacted their care processes negatively. Impacts of oral diseases included pain, loss of sleep, and dismissal from work. Those who did receive care appreciated the quality and scope of treatment. Suggestions for improvements to the current services and policies include the following: a more inclusive healthcare policy; lower fees; integration of dental care into public insurance; and creation of community dental clinics [22].

Work to accomplish: Validate the results of Phase 1a with focus group study (see details below).

## 2. Methods and Analysis

### 2.1. Study Design

This project uses a sequential mixed methods design [23] as illustrated in Figure 1. A mixed methods design answers both “what?” and “how?” questions and enables the uncovering of the complexity of health experiences and behaviours [24]. In concurrent design, the qualitative and quantitative components are independent and the data of each strand are analyzed separately. The results are then merged to build interpretations and draw conclusions [23]. The mixed methods design is important in applied dental public health, as it enables exploring the phenomenon from multiple perspectives, contributing to an in-depth understanding of oral health issues [25].

This project has three phases: Phase 1a (Qualitative; described above) for RQ1 and RQ2; Phase 1b (Quantitative) corresponds to RQ3 and RQ4; Phase 2 (Mixed Analysis); and Phase 3 (Knowledge Translation and dissemination).

#### 2.1.1. Phase 1a, Focus Group Verification

To enhance the transferability of our qualitative study, we will conduct focus groups with humanitarian migrants in the province of Ontario, basing our interview guide on our results from the qualitative study completed in the province of Quebec. We will create composite exemplars of common and extreme participant experiences to encourage discussion. We will aim for focus groups in our five collaborating centres. Using best practices for focus groups [26], we will separate groups by gender and language, seeking six to eight participants per group. Two members of our team (NN and MK) with expertise in qualitative research who speak English, French, and Arabic will facilitate the focus groups in the language of the participants. We will train an interpreter to facilitate any focus group held in other languages, if necessary. Data from focus group discussions (~90 min) will be audio-recorded then translated into English at the point of preliminary analysis in the event of team members not having the specific linguistic ability. Analysis will be thematic, seeking concordance and discordance with completed Phase 1a (Qualitative) results.

#### 2.1.2. Phase 1b, Clinical Examination and Quantitative Survey

For RQ3, we will measure the prevalence of dental caries, periodontal disease, oral pain, and traumatic dental injuries, comparing results with CHMS across variables (age, gender, family income, education, born inside or outside Canada). We hypothesize that a significantly higher proportion of humanitarian migrants will have untreated dental caries compared to Canadians. For RQ4, we will describe the factors that limit access to oral health care and analyze for predictors, using the survey data.

### 2.2. Study Population

Over two-thirds of humanitarian migrants arrive in Canada via Quebec and Ontario [2], thus we will recruit in these provinces. *Inclusion criteria:* In order to be included, participants must adhere to the following criteria: currently be, or have been, a refugee or asylum seeker; age 18–79 years (as per upper age limit of CHMS); have lived anywhere in Canada <2 years (to cover the critical periods of adaptation to host country while ensuring the ‘healthy immigrant effect’ [27] is not yet advanced); be able to communicate in main study languages (English, French, Arabic, Spanish); provide verbal or written consent; agree to complete both clinical exam and questionnaire. *Exclusion criteria:* We will exclude people with oral heath restrictions as per the CHMS criteria (e.g., people who require antibiotics before dental appointments, have been diagnosed with heart murmurs, or had joint replacement).

### 2.3. Sampling

The methodological challenges involved in research with migrants and other minorities are known [28,29]. Further, the demographic profile of humanitarian migrants in Canada is in flux. While official statistics can accurately indicate variables (e.g., from what country a person applied for asylum), we cannot track where humanitarian migrants are currently living once they arrive in the country. In addition, our target population is vulnerable and may be difficult to recruit. Such challenges make it difficult to ensure a representative sample. Thus, we will use Respondent-Driven Sampling (RDS) [28,30], a technique that combines ‘snowball sampling’ with a mathematical model that weights the sample to compensate for the non-random approach [31]. RDS is increasingly popular for hard-to-reach populations [28], building upon trust already embedded in communities. Our team has developed such trust.

The first step of RDS is to recruit front-line participants, called ‘seeds’. Seeds then recruit up to three other participants, who then can recruit up to three participants, etcetera. We will begin with three seeds in each of five community organizations. Seeds will be given three coupons to give to relevant participants. Each uniquely numbered coupon will have a brief summary of the project, contact information, and an expiration date (3 months, to facilitate recruitment). Participants will be offered monetary compensation for their time ($20), and an incentive for each new participant they recruit ($5). The number of participants from each organization will take into account the size of the target population.

### 2.4. Recruitment

We will create study pamphlets and posters in lay English, French, Arabic, and Spanish (participant languages) and prepare electronic messages for LISTSERVS and social media. We will seek permission to post flyers at popular community sites and social media and websites. Our staff will attend community events, such as orientation sessions for newly arrived refugees and immigrants, to introduce the study. Study information will include how to contact our staff directly or via the organisation. We will contact interested participants and explain the purpose of the study to gauge willingness to participate and fit. If they agree, we will schedule an appointment at the time and place of their choice. On the day of data collection, we will collect coupons from potential participants and assess their eligibility for the clinical exam. The complete information about the project will be given, and participants will be invited to participate, and sign, or verbally consent to, the protocol. 

After consenting, we will ask eight questions to gather essential information to compute an unbiased estimate of population parameters, including the following examples: (i) Approximately how many humanitarian migrants do you know personally?; (ii) Of those, how many do you know by name and have their contact information?; (iii) Of those, how many may also know you?; (iv) Of those, how many have you contacted in the last six months? Responses will be used to calculate the personal network size of each participant for RDS estimation weights. Reasons for non-participation and basic descriptive data will be recorded with permission. Coupons from non-eligible and nonparticipating individuals will be retained.

Note: we will also request permission to re-contact participants in the future in anticipation of a longitudinal study.

### 2.5. Sample Size

Studies using RDS require a complex sampling design. Our sample size calculation was based on RQ3 and follows Cornfield’s recommendation to account for this design effect [32]. Figure 2 shows different scenarios for the sample size calculation.

We will require 420 individuals to estimate a 30% prevalence of untreated caries with a margin of error of 0.08 from a population of 140,000 humanitarian migrants in the study regions (Ontario and Quebec), and a response rate of 60%. Prevalence of untreated caries in the CHMS was 20%. We expect a higher prevalence of untreated caries among humanitarian immigrants.

### 2.6. Ethics and Dissemination

The study has been approved by the McGill Institutional Review Board (IRB Study Number A06-B24-18A.) and the Research and Ethics Board of the University of Toronto (RIS Protocol Number: 36911). Collaborating community organizations and knowledge users provided letters that confirmed their willingness to participate in the research project. Written consent will be requested from participants; verbal consent will be accepted. Participation in this research project is a voluntary decision. Participants can decide not to answer any question they are uncomfortable with and withdraw from the study at any time without negative consequences. In the case of withdrawal, they can choose to have any clinical or questionnaire data collected to that point destroyed (note: focus group data cannot be disaggregated). Participants will be compensated for time and transportation. In the event of dental emergency, we will endeavour to find inexpensive care; participants needing follow-up will be referred to clinical facilities. Some participants may have survived torture and trauma; all team members will attend sensitivity training by one of our team members trained in psychology, including referral sources.

We will ensure that any information provided by participants is used strictly for the purpose of this study. Data will be anonymized, and electronic versions stored on a secure McGill University research server. Hard copies of data will be stored in a secure cabinet, behind a locked door, either at McGill University or the University of Toronto. Only persons directly involved in the study will be able to access the secure data; however, members of the university ethics boards may access the study data to verify the ethical and responsible conduct of the study. Data will be destroyed seven years after publication; electronic data will be deleted from the server and hard copies of data will be shredded.

We will hold initial team meetings to ensure data of interest to communities will be captured. Annual meetings will be organized to discuss progress, and ongoing communication will be developed as desired by study sites. The Phase 3 meeting will help to interpret results and design next steps. We will produce an end-of-grant executive summary and ‘fact sheet’ for wide distribution (e.g., to educational, clinical, and policy stakeholders). Sensitizing workshops with clinicians and students, and roundtable discussions (e.g., Café Scientifique-style assemblies) to disseminate results in lay contexts and brainstorm ways forward are anticipated. Project results will be disseminated via LISTSERVs and bulletins, conference abstracts, oral presentations, and publications in peer-reviewed journals.

## 3. Data Collection

### 3.1. Oral Health Status

Oral health status will be measured with four clinical parameters: dental caries, periodontal disease, traumatic dental injury (TDI), and oral pain. Detailed definition of the first three, their categorisation of severity, and the coding system, can be found in the CHMS Dentist’s Survey Manual and Coding Criteria [19]. We will supplement the questions on pain with two components of the McGill Pain Questionnaire (short form): [33]. The Present Pain Intensity scale (PPI) (0—no pain; 1—mild; 2—discomforting; 3—distressing; 4—horrible; 5—excruciating) and the accompanying Visual Analog Scale (VAS) (100 mm, no pain to worst possible pain). Data will be collected during a full oral health exam, after which participants will be given treatment recommendations, oral hygiene supplies, and referral, if required, as per the CHMS protocol. The clinical exam takes approximately 13 min [19]; the VAS and PPI will take less than 5 min, total [33]. Participants will be invited to the Faculty of Dentistry and the satellite clinics for data collection. The clinical examination will be carried out using a mouth mirror and calibrated periodontal probe; we will not use x-rays.

### 3.2. Structured Questionnaire

Our questionnaire combines two national surveys: the CHMS-OHC household survey [19] and CCHS-Optional set with research literature that captures OHL [20] and migrant experience [12,29]. It will be translated and validated so that it can be administered by a research assistant in the participants’ language of choice. It seeks information on the following factors: socio-demographic information (e.g., age, gender, country of origin, migrant status, time in refugee camp, religion, length of time in Canada, education, income); general health; diet; smoking; soft drink consumption; satisfaction with oral health (e.g., appearance, stigma); oral symptoms (including pain); dental care habits (e.g., frequency of tooth brushing, visits to a dental professional, non-Western hygiene practices); OHL; and barriers to dental care (e.g., fear, finances, language). It takes approximately 30 min to complete and will precede the clinical examination.

### 3.3. Data Quality

Data quality will be ensured through standardized data collection procedures. Once tools are translated, they will be reviewed for cultural acceptability, burden, and linguistic accuracy [34]. Our team includes a principal trainer for the CHMS-Oral Health Component (OHC) and a clinician calibrated to WHO standards as a CHMS-OHC examiner; they will train our staff. Data collection will begin with simulations to ensure standardization across participants.

### 3.4. Data Analysis

**Statistical analyses:** Exploratory data analysis will be conducted to identify the main patterns, average values, dispersion, distribution shape, and presence of outliers for each variable. For RQ3, participants’ oral health status will be measured as follows:

Caries experience (a dental term for decayed, missing, and filled teeth as a result of caries), periodontal disease and traumatic dental injuries will be expressed as the proportion of participants with decayed, missing, and filled teeth (DMFT > 0), presence of untreated Decay (D > 0), Loss of Attachment (LOA ≥ 3 mm), and the presence of traumatic dental injuries, respectively.

The severity of caries experience and periodontal disease will be measured by the mean DMFT > 6.85 and the mean proportion of sites with LOA > 6 mm.

Treatment needs will be assessed by the presence of untreated decay (D > 0) and LOA above 6 mm.

The prevalence of different aspects of oral pain will be described using the proportion of subjects who have had in the last month: “a toothache,” “pain in their teeth when consuming hot or cold foods or drinks,” “severe tooth or mouth pain at night,” “pain in or around jaw joints” or “other pain in your mouth.” Frequency of global oral pain experience will be determined by the proportion of participants who have never experienced oral pain or who have experienced it rarely, sometimes, or often in the past month. We will characterize current oral pain intensity by calculating the mean and standard deviation of PPI and VAS scores.

We will use χ2 (categorical variables) and *t*-tests (continuous variables) to compare the frequency distributions of each indicator of oral health status according to age, gender, income, education, and country of origin. We will estimate the associations of socio-demographic and behavioural characteristics and our primary outcome variable (presence of untreated decay) using unconditional logistic regression and 95% confidence intervals (CIs). We will compare the frequency distributions of each indicator of oral health status of our survey with those from the CHMS.

For RQ4, barriers to oral health care will be conceptualized via the BMVP and will be measured using questions addressing cost, insurance, transportation, mobility, health, awareness of resources, language, stigma, fear, time, and availability. The outcome—access to oral health care—will be measured as a visit to the dentist in the past year. We will use hierarchical models to select the main determinants of access to oral health care. The goodness of fit of the models will be assessed using the Akaike information criterion [35]. We will use imputation techniques to deal with missing values.

### 3.5. Phase 2, Mixed Methods Integration

Phase 2 will answer the overarching research question *What is the oral health status of newly arrived humanitarian migrants and how can oral health services better meet their needs?* After each data set (qualitative, quantitative) is analyzed independently, analyses will be integrated following Creswell [23]. We will develop a matrix, as shown in Table 1, linking data with the BMVP.

The matrix will seek convergence, complementarity, uniqueness, and discrepancy [36]. For example, our examination of ‘barriers’ in the quantitative analysis may both overlap with, and diverge from, the inductive qualitative analysis. We can anticipate a range of possible barriers (e.g., sociocultural: ethnic conceptions of health, aesthetics; psychological: trauma; experiential: discrimination; structural: finances) and how they operate socially (e.g., stigmatization).

### 3.6. Phase 3, Knowledge Translation and Future Directions

Finally, a knowledge translation (KT) meeting will be held in Montreal with all team members. We will provide data summaries and work towards in-depth explanations of all relationships in the matrix [23]. ‘Knowledge users,’ meaning individuals who are likely to make use of the research results to inform their decisions and practices (e.g., The Office of the Chief Dental Officer, leaders of community organizations) will help refine interpretations. Together, we will advance areas for policy and practice changes and design health promotion interventions to be developed for future funding.

## 4. Expertise, Activities, Timeline

Our team is multilingual (12 languages) and interprofessional, with expertise in epidemiology, social sciences, clinical dentistry, ethics, migrant health, psychology, and nursing. We have research experience in health services, determinants of health, health policy, public health, and dental public health. We have advanced methodological training in qualitative, quantitative, and mixed methods research. Our collaborators and knowledge users include national and provincial professional organizations, managers, and academic and community leaders.

## 5. Strengths and Limitations of This Study

Our research design allows for the collection of data from multiple sources using a variety of approaches. The triangulation of data sources and techniques will contribute to the richness of the data and analysis. Our team is multilingual (12 languages) and interprofessional, with expertise in dental public health, epidemiology, social sciences, clinical dentistry, ethics, migrant health, psychology, and nursing. The research design is built upon community and clinic collaborations; these relationships will foster and enhance knowledge translation to community and clinical areas, ultimately optimizing the care of humanitarian migrants.

As with any survey, respondents may be subject to recall bias. The clinical exam may help trigger memories; however, it may also trigger unpleasant memories for which trained professionals are available to manage. Some newly arrived migrants may be reluctant to participate in research (e.g., out of fear of deportation); however, this has not been experienced to date. It is anticipated that the free clinical exam will enhance recruitment. The participants will be provided travel compensation, oral hygiene instructions, and free dental supplies.

## 6. Significance

This project will: (1) provide benchmarking data on the oral health status of humanitarian migrants; (2) advance understandings of the sociocultural context within which migrants experience oral health and service delivery; (3) advance understandings of how the current oral health system could be improved; (4) make available findings to local, national, and international communities; (5) increase trainees’ expertise in migrant health; (6) build collaborations ensuring KT to community and clinical areas to optimize the health of humanitarian migrants; (7) enable a national program of longitudinal research to improve the oral health of humanitarian migrants in Canada.

## Figures and Tables

**Figure 1 ijerph-16-00542-f001:**
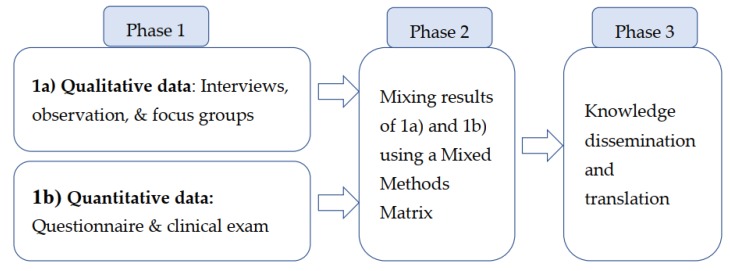
Diagram of mixed method study design.

**Figure 2 ijerph-16-00542-f002:**
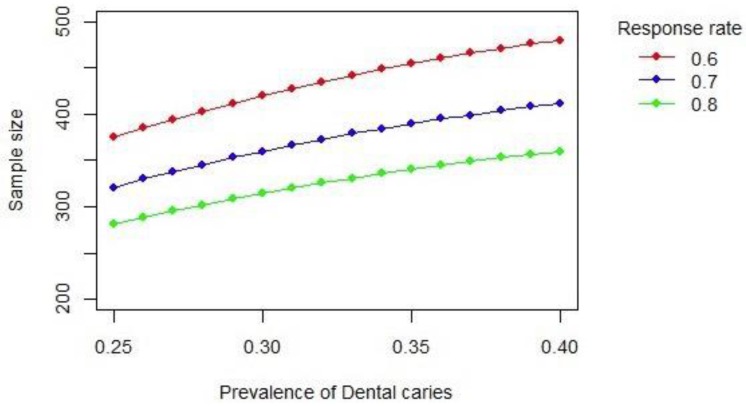
Scenarios for the sample size calculation.

**Table 1 ijerph-16-00542-t001:** Example of matrix for integrating and interpreting results. (Note: this combines actual and hypothetical data).

Interpretation of ‘Barriers’		
Using data from: Phase 1a (Qualitative); RQ1 and RQ2 Deductive components (“If you went to see a helper or healer of any kind, please tell me about your visit and what happened afterwards.” + Any challenges/barriers (probe for the Behavioral Model for Vulnerable Populations variables: stigma/fear, finances/insurance, linguistic challenges, transportation, mobility/health, awareness of resources, and time/availability). Inductive emergent themes. Phase 1b (Quantitative); RQ4: Analysis that maps to components of Behavioral Model for Vulnerable Populations: cost/insurance, transportation, mobility/health, awareness of resources, language, stigma/fear, and time/availability		
**What** = identification of what barriers exist. **How** = examining how barriers operate.		
Relevant actual qualitative results **(How)**	Relevant hypothetical quantitative results **(What)**	Hypothetical integration: Convergence, complementarity, divergence, uniqueness
Low importance of oral health in cultures of origin; yet, participants were aware of the importance of good oral health to their wellbeing	n/a	Unique to qualitative: Qualitative data highlights how culturally specific views affect care seeking. Important for culturally attuned service development. Could be used in future survey research?
Lack of finances and lack of dental insurance were the main issues that prevented participants from seeking dental care.	Lack of finances/insurance associated with care avoidance	Convergence and uniqueness: Quantitative data suggests a possible connection between financial factors and the social process of stigmatization, both of which impede access to oral health care. Could follow up with this in the verification focus groups.
	Stigma/fear associated with care avoidance	
Oral health was a priority for participants, possibly resulting from disease experience	Lack of time/availability associated with low care seeking	Divergence; to discuss with larger team
Oral disease impacts were prevalent and limited daily functions of participants	Mobility/health keeps some people from seeking care	Convergence; this seems especially important for service planning.
Participants felt community organizations were useful for finding a dentist who would accept to provide care despite their financial limitations	Lack of awareness of resources represents a common reason for not seeking care.	Divergence; Hypothesis: Perhaps sampling bias in qualitative sample?
Observations revealed that language constituted a barrier during episodes of care	Language prevents people from seeking care	Convergence (language) and uniqueness (race)
Overall, participants felt they could use public transport systems well	Transportation represents a common reason for avoiding care	Divergence (return to data to re-assess)

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
