# Peer review of "The Oral Health of Refugees and Asylum Seekers in Canada: A Mixed Methods Study Protocol"

_ijerph, 2019, doi:10.3390/ijerph16040542_

Round 1
Reviewer 1 Report
This is a study protocol want to investigate the oral health of refugees and asylum seekers in Canada and build a long-term oral health care programme for this population. There are several aspects that the authors can consider improving this manuscript. First, I don’t think the manuscript is well prepared. Please check the format of the manuscript. A lot of minor issues should be addressed.
Title:
1. From the title, I will consider this study protocol is about the oral health of refugees and asylum seekers in Canada. However, the ultimate aim of this protocol is to build a long-term oral health care programme for this population. Therefore, I think the title is not appropriate.
Abstract:
1. Rather than state the number of refugees and asylum seekers globally, the number of this group of people in Canada is more relevant to this study.
2. Please specify this is a ‘study protocol’ in the abstract.
Introduction:
1. Line 50, what did the authors mean by ‘health… decrease…’?
2. The authors should state clearly that this paper is a study protocol.
Methods:
1. If I am correct, the phase 1a consists of two parts: 1) interview to single immigrant and health workers, 2) focus group discussion. Please clarify this part in 2.1.1.
2. Please provide information of the interviewers. Any training session or qualification?
3. Details of how the qualitative data is recorded, transferred, interpreted, etc. should be included.
4. I have a question about why the authors conducted the qualitative part prior to the quantitative part. To my understanding, if oral health data can be collected first, the interviews and focus group discussion can be more specific and in depth by referring to the oral health situations of this group of people. Please illustrate.
5. The clinical data can be significant different between different age groups. In the manuscript, the authors haven’t provided any information about the age group of the participants. Please provide the information and discuss.
Data collection:
1. Please provide the information about the venues and material used for clinical data collection.
Author Response
Reviewer #1
Title:
i) From the title, I will consider this study protocol is about the oral health of refugees and asylum seekers in Canada. However, the ultimate aim of this protocol is to build a long-term oral health care programme for this population. Therefore, I think the title is not appropriate.
We understand the concern of the reviewer. However, we do not feel comfortable modifying the title at this stage as it has passed through various scientific bodies: The CIHR funding committee, the McGill University Institutional Review Board, and University of Toronto Ethics and Review Board.
Abstract:
ii) Rather than state the number of refugees and asylum seekers globally, the number of this group of people in Canada is more relevant to this study.
We agree with this remark and have revised the opening sentence (Abstract: Line 14).
iii) Please specify this is a ‘study protocol’ in the abstract.
We have added the word “protocol” in the abstract (Abstract: Line 15).
Introduction:
iv) Line 50, what did the authors mean by ‘health… decrease…’?
We have replaced the word “decrease” with “declined” (Introduction: line 54)
v) The authors should state clearly that this paper is a study protocol.
We have included the word “protocol” as suggested (Introduction: Line 61).
Methods:
vi) If I am correct, the phase 1a consists of two parts: 1) interview to single immigrant and health workers, 2) focus group discussion. Please clarify this part in 2.1.1.
We have expounded on 2.1.1 to clarify the concern raised (Methods & Analysis: Lines 144-145).
vii) Please provide information of the interviewers. Any training session or qualification?
We have provided basic information about the interviewers to respond to this concern (Methods and Analysis: Lines 148-150).
viii) Details of how the qualitative data is recorded, transferred, interpreted, etc. should be included.
We have revised the sentence on data collection to address the above concern (Methods and Analysis: Lines 151-152).
ix) I have a question about why the authors conducted the qualitative part prior to the quantitative part. To my understanding, if oral health data can be collected first, the interviews and focus group discussion can be more specific and in depth by referring to the oral health situations of this group of people. Please illustrate.
In our original conception of the project, we intended to follow this logic. Our funding opportunity, however, did not support this possibility. Our first grant only enabled us to complete the qualitative component. Our current grant is more substantial, and allows us to complete the entire mixed-methods study.
x) The clinical data can be significant different between different age groups. In the manuscript, the authors haven’t provided any information about the age group of the participants. Please provide the information and discuss.
We agree with the remark that clinical data can be significantly different between two age groups. In this protocol, we defined the inclusion criteria (Study Population: Lines 164-167) and mentioned that we will compare our results to that of the CHMS which does include sub-group analysis by age (Data Analysis: Lines 309-310).
Data collection:
xi) Please provide the information about the venues and material used for clinical data collection.
We have provided further information (Data collection. Lines 265-267).
Reviewer 2 Report
abstract
What is the aim of the study? to assess the situation globally or in Canada 'The purpose of this project is to generate robust baseline data on the oral health of this population and build a long-term program of research to improve their access to dental care in Canada.' rewrite these sentences as you referring to previous sentence (global) but at the end you mention Canada, the background should be focused on Canada, not numbers in global!
Rewrite the following and add more info for general reader' sequential mixed methods design, with the Behavioral Model for Vulnerable Populations as the conceptual framework.' again later on' mixed methods interpretation'
Mention for the reader that 'Over two-thirds of humanitarian migrants arrive in Canada via Quebec and Ontario, thus,you recruit in these provinces.' add also the eligibility and exclusion criteria for study participants
Define the 'stakeholders' in the last part of the abstract
methods
Did you do any sample size calculation to come up with '420 humanitarian migrants in Ontario and Quebec' add more detail, later on you mention that this is based on '30% prevalence of untreated caries' and also' Prevalence of untreated caries in the CHMS was 20%. We expect a higher prevalence of untreated caries among humanitarian immigrants.' add referece for this 20% figure
In your research question you came up with RQ1-RQ4, did you do any pilot study and on that bases you identified these areas? or you based this on previous reseach in canada?
Is this based on the study you mentioned later on' sample of 25 humanitarian migrants and 10 healthcare providers in Montreal, Quebec'
Can you add the copy of the Questionnaire that you want to use for your interview ,are these Questionnaires vaidated for the specific population you want to use for?
please add a table with variables (dental non-dental) that you want to collect
Author Response
Reviewer 2
Abstract
i) What is the aim of the study? to assess the situation globally or in Canada 'The purpose of this project is to generate robust baseline data on the oral health of this population and build a long-term program of research to improve their access to dental care in Canada.' rewrite these sentences as you referring to previous sentence (global) but at the end you mention Canada, the background should be focused on Canada, not numbers in global!
We agree with this remark and have revised the opening sentence (Abstract: Line 14).
ii) Rewrite the following and add more info for general reader' sequential mixed methods design, with the Behavioral Model for Vulnerable Populations as the conceptual framework.' again later on' mixed methods interpretation'
We have revised the methodology section for clarity. (Abstract: Lines 24-26). We are unable to expand on the methodology due to the word limit of the abstract.
iii) Mention for the reader that 'Over two-thirds of humanitarian migrants arrive in Canada via Quebec and Ontario, thus, you recruit in these provinces.' add also the eligibility and exclusion criteria for study participants
We are unable to integrate this useful suggestion due to the word limit of the abstract. Please advise.
iv) Define the 'stakeholders' in the last part of the abstract
We have listed some of the stakeholders as suggested (Abstract: Lines 26-27).
Method
v) Did you do any sample size calculation to come up with '420 humanitarian migrants in Ontario and Quebec' add more detail, later on you mention that this is based on '30% prevalence of untreated caries' and also' Prevalence of untreated caries in the CHMS was 20%. We expect a higher prevalence of untreated caries among humanitarian immigrants.' add reference for this 20% figure
Yes, we explained that our sample size calculation was based on an estimated prevalence of 30% in the study population. Figure 2 illustrates the possible sample sized based on our recruitment approach and response rates.
vi) In your research question you came up with RQ1-RQ4, did you do any pilot study and on that bases you identified these areas? or you based this on previous research in Canada? Is this based on the study you mentioned later on' sample of 25 humanitarian migrants and 10 healthcare providers in Montreal, Quebec'
Our research program builds upon findings of a scoping review “The oral health of refugees and asylum seekers: a scoping review. Globalization and Health. 2016, 12, 59.” The qualitative study with humanitarian migrants was the first step of our empirical study.
vii) Can you add the copy of the Questionnaire that you want to use for your interview, are these questionnaires validated for the specific population you want to use for?
We mentioned that the questionnaire will be translated and reviewed for cultural acceptability, burden and linguistic accuracy (Data quality Lines: 281-283). The validated CHMS questionnaire is available at: http://www23.statcan.gc.ca/imdb-bmdi/pub/document/5071_D2_T1_V1-eng.htm.
viii) Please add a table with variables (dental non-dental) that you want to collect
Details of the variables of interest are provided in the above-referenced CHMS document. Please advise if you would still like an additional table.
Reviewer 3 Report
It is a benefit to the Canadian healthcare system to understand the oral health status of this population.
Look forward to the actual data.
Author Response
Reviewer 3
It is a benefit to the Canadian healthcare system to understand the oral health status of this population.
Look forward to the actual data.
We thank the reviewer for the positive feedback.